# DISENTANGLING DATA DISTRIBUTION FOR FEDERATED LEARNING

## ABSTRACT

Federated Learning (FL) facilitates collaborative training of a global model whose performance is boosted by private data owned by distributed clients, without compromising data privacy. Yet the wide applicability of FL is hindered by entanglement of data distributions across different clients. This paper demonstrates for the first time that by disentangling data distributions FL can in principle achieve efficiencies comparable to those of distributed systems, requiring only one round of communication. To this end, we propose a novel FedDistr algorithm, which employs stable diffusion models to decouple and recover data distributions. Empirical results on the CIFAR100 and DomainNet datasets show that FedDistr significantly enhances model utility and efficiency in both disentangled and near-disentangled scenarios while ensuring privacy, outperforming traditional federated learning methods.

## 1 INTRODUCTION

Despite the extensive research in Federated Learning (FL), its practical application remains limited. A key challenge is to achieve high efficiency while preserving both model utility and privacy (McMahan et al., 2017; Kairouz et al., 2021). There is a consensus that this inefficiency stems from the *entanglement* of data distribution across clients, where many rounds of communications are required to ensure the convergence of the global model (Zhao et al., 2018; Li et al.; Tian et al., 2024). The opposite of *entanglement* implies that client data distributions at different dimensions are disentangled as shown in Figure 1(a).

We believe that the ideal federated learning algorithm can achieve efficiency levels comparable to those of ideal distributed systems that attain full parallelism (Sunderam, 1990), provided that the data distributions across clients can be entirely *disentangled*, as illustrated in Fig. 1(a) (see details in Def. 2). As shown by Theorem 1, it is actually a

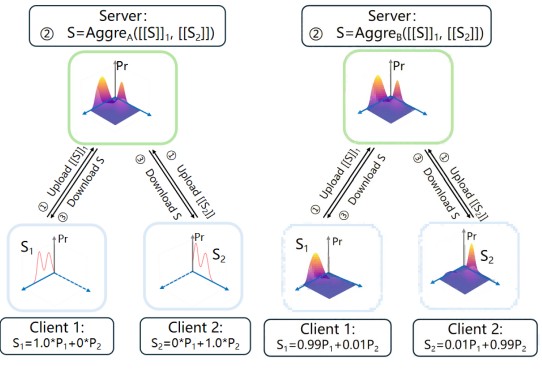

(a) Disentangled      (b) near-disentangled

**Figure 1:** Disentangled and near-disentangled cases: In the Disentangled case, two clients have data distributions on two disentangled base distribution, $P_1$ and $P_2$, separately. In the $\xi$-entangled case, each client has data distributions across both disentangled base distribution $P_1$ and $P_2$, but with one base distribution dominating the other. In both case, client $k$ ($k = 1, 2$) communicates with the server through data distribution in a single round (upload the distribution $[[S_k]]$ that applying the privacy preserving mechanishm on $S_k$). The server employs different aggregation strategies for the two scenarios: $\text{Aggre}_A$ and $\text{Aggre}_B$ for disentangled and near-disentangled data distributions respectively.

*sufficient condition* for achieving ideal efficiency of federated learning with only one round of communication being needed. Moreover, this ideal condition is often approximately fulfilled in practice e.g., when millions of mobile device clients participate in federated learning and most client data distributions are different from each other (Sattler et al., 2019; Tian et al., 2024). In other words, under such a near-disentangled condition (see Def. 2 and Fig. 1(b)), there exists a federated learning algorithm capable of achieving global model utility with only one communication round, while ensuring that the utility loss remains within a tolerable threshold (see Theorem 2).

In order to take full advantage of the described disentangled and near–disentangled cases (see Fig. 1), we propose to first disentangle data distributions into distinct components such that each client can launch their respective learning task independently. This disentanglement will lead to significant reduction of required communication rounds as revealed by Theorem 1. Technology-wise, there is a large variety of methods that can be utilized to decompose data distribution, ranging from *subspace decomposition* (Abdi & Williams, 2010; Von Luxburg, 2007) to *dictionary learning* (Tošić & Frossard, 2011) etc. (see Section 2.2 for detailed review). In this work, we propose an algorithm, called *FedDistr*, which leverages the stable diffusion model technique (Croitoru et al., 2023) due to its robust ability to extract and generate data distributions effectively. In our approach, clients locally extract data distributions via stable diffusion model, and then upload these decoupled distributions to the server. The server actively identifies the orthogonal or parallel between the base distributions uploaded by clients and aggregate the orthogonal distribution once.

Previous work (Zhang et al., 2022) has shown that it is impossible to achieve the optimal results among the utility, privacy and efficiency. The proposed FedDistr offers a superior balance between model utility and efficiency under the disentangled and near-disentangled scenarios, while still ensuring privacy-preserving federated learning: (1) By decoupling client data distributions into the different base distributions, the server actively aligns the base distribution across different client, while FedAvg does not perform this decoupling, leading to a decline in global performance, especially for the disentangled case; (2) The proposed FedDistr requires only one round of communication, and the amount of transmitted distribution parameters is much smaller than that of model gradients; (3) The proposed FedDistr transmits a minimal amount of data distribution parameters, thereby mitigating the risk of individual data privacy leakage to some extent. Furthermore, privacy mechanisms such as differential privacy (DP) can be integrated into FedDistr, offering additional protection for data privacy.

## 2 RELATED WORK

### 2.1 COMMUNICATION EFFICIENT FEDERATED LEARNING

One of the earliest approaches to reducing communication overhead is the FedAvg algorithm (McMahan et al., 2017). FedAvg enables multiple local updates at each client before averaging the models at the server, thereby significantly decreasing the communication frequency. To further mitigate communication costs, compression techniques such as quantization and sparsification (Reisizadeh et al., 2020) and client selection strategies (Lian et al., 2017; Liu et al., 2023) have been implemented in federated learning. However, all of these methods are less effective in Non-IID scenarios, where the model accuracy is significantly impacted.

To address the Non-IID problem, numerous methods have been proposed (Li et al., 2020b; Arivazhagan et al., 2019), which introduce constraints on local training to ensure that models trained on heterogeneous data do not diverge excessively from the global model. However, most of these methods require numerous communication rounds, which is impractical, particularly in wide-area network settings.

Furthermore, some methods propose a one-shot federated learning approach (Guha et al., 2019; Li et al., 2020a), which requires only a single communication round by leveraging techniques such as knowledge distillation or consistent voting. However, significant challenges arise in this context under Non-IID settings (Diao et al., 2023).

### 2.2 DISTRIBUTION GENERATION

This paper focus on transferring distribution instead of models. There are several distribution generation methods: **Parametric Methods** (Reynolds et al., 2009) assume that the data follows a specific distribution with parameters that can be estimated from the data. *Gaussian Mixture Model (GMM)* represents a distribution as a weighted sum of multiple Gaussian components. **Generative Models** (Goodfellow et al., 2020; Croitoru et al., 2023) aim to learn the underlying distribution of the data so that new samples can be generated from the learned distribution. For example, GANs Goodfellow et al. (2020) consist of two networks, a generator $G$ and a discriminator $D$. The generator learns to produce data similar to the training data, while the discriminator tries to distinguish between real and generated data. **Principal Component Analysis (PCA)** (Abdi & Williams, 2010) is a linear

method for reducing dimensionality and capturing the directions of maximum variance. It decomposes the data covariance matrix $\Sigma$ into eigenvectors and eigenvalues. **Dictionary Learning and Sparse Coding** (Tošić & Frossard, 2011) seeks to represent data as a sparse linear combination of basis vectors (dictionary atoms).

Table 1: Table of Notation

| Notation | Meaning |
|---|---|
| $\mathcal{D}_k$ | Dataset of Client $k$ |
| $K$ | the number of clients |
| $\mathcal{D}$ | $\mathcal{D}_1 \cup \cdot \cup \mathcal{D}_K$ |
| $S_k$ and $S$ | the distribution of $\mathcal{D}_k$ and $\mathcal{D}$ |
| $\epsilon_u$ | Utility loss in Eq. (4) |
| $F_\omega$ | the federated model |
| $\{P_i\}_{i=1}^m$ | $m$ independent sub-distributions |
| $\vec{\pi}_k = (\pi_{1,k}, \cdots, \pi_{m,k})$ | the probability vector for client $k$ on $\{P_i\}_{i=1}^m$ |
| $\xi$ | entangled coefficient |
| $s_{k_1,k_2}$ | Entangled coefficient between client $k_1$ and $k_2$ |
| $p$ | learnable prompt embedding |
| $T_c$ | communication rounds |

## 3 FRAMEWORK

In this section, we first formulate the problem via distribution transferring and then provide the analysis on what conditions clients can directly communicate once through a distribution entanglement perspective.

### 3.1 PROBLEM FORMULATION

Consider a Horizontal Federated Learning (HFL) consisting of $K$ clients who collaboratively train a HFL model $\omega$ to optimize the following objective:

$$\min_\omega \sum_{k=1}^{K} \sum_{i=1}^{n_k} \frac{f(\omega; (x_{k,i}, y_{k,i}))}{n_1 + \cdots + n_K}, \tag{1}$$

where $f$ is the loss, e.g., the cross-entropy loss, $\mathcal{D}_k = \{(x_{k,i}, y_{k,i})\}_{i=1}^{n_k}$ is the dataset with size $n_k$ owned by client $k$. Denote $\mathcal{D} = \mathcal{D}_1 \cup \cdots \cup \mathcal{D}_K$ and $\mathcal{D}$ follows the distribution $S$. The goal of the federated learning is to approach the centralized training with data $\mathcal{D}$ as:

$$\min_\omega \mathbb{E}_{(x,y) \in S} f(\omega; (x, y)). \tag{2}$$

Numerous studies (Zhao et al., 2018; Li et al.; Tian et al., 2024) have demonstrated that Eq. (1) converges to Eq. (2) under IID settings. However, significant discrepancies arise in Non-IID scenarios, leading to an increased number of communication rounds and a decline in model utility for Eq. (1). While data heterogeneity in extreme Non-IID settings was considered the main cause for FL algorithms having to compromise model performances for reduced rounds of communication, this paper discloses that direct optimization of Eq. (2) can actually be achieved at the cost of a single round of communication. Both theoretical analysis and empirical studies show that data heterogeneity is a blessing rather than a curse, as long as data distributions among different clients can be completely disentangled (see Theorem 1). Moreover, we aim to learn the distribution $S$ in a single communication by transferring either the distribution or its parameters, thereby mitigating the risk of individual data privacy leakage (Xiao & Devadas, 2023) as follows:

$$\min_S \sum_{k=1}^{K} \lambda_k \tilde{f}(S; \mathcal{D}_k), \quad \text{s.t. } T_c = 1 \tag{3}$$

where $\tilde{f}$ represents the loss function used to evaluate whether $S$ accurately describes the dataset $\mathcal{D}_k$ [1], $T_c$ is the communication round, $\lambda_k$ denotes the coefficient for client $k$, and it is required that $\sum_{k=1}^{K} \lambda_k = 1$.

In this framework, all clients first collaborate to estimate the distribution $S$ of the total dataset $\mathcal{D} = \mathcal{D}_1 \cup \cdots \cup \mathcal{D}_K$. The following section discusses the conditions under which clients can directly interact once to estimate $S$ through a distribution entangled perspective.

We also define the utility loss for the estimated distribution $\hat{S}$ as:

$$\epsilon_u = \mathbb{E}_{(x,y) \in S} f(\hat{\omega}^*; (x,y)) - \mathbb{E}_{(x,y) \in S} f(\omega^*; (x,y)), \tag{4}$$

where $\omega^*$ is the optimal parameter of Eq. (2) and $\hat{\omega}^* = argmin_{\omega} \mathbb{E}_{(x,y) \in \hat{S}} f(\omega^*; (x,y))$.

**Threat Model.** We assume the server might be *semi-honest* adversaries such that they do not submit any malformed messages but may launch *privacy attacks* on exchanged information from other clients to infer clients' private data.

### 3.2 DISTRIBUTION ENTANGLEMENT ANALYSIS

#### 3.2.1 $\xi$-ENTANGLED

Assume the data $\mathcal{D}$ is drawn from $m$ distinct base distributions. Therefore, we can decompose the complex distribution $S$ into simpler components, each representing a portion of the overall distribution (Reynolds et al., 2009; Abdi & Williams, 2010):

$$S = \sum_{i=1}^{m} \pi_i P_i. \tag{5}$$

where $P_i$ represents the $i$-th base distribution (e.g., DomaiNet has the data with different label and domain, the base distribution represents the distribution followed by

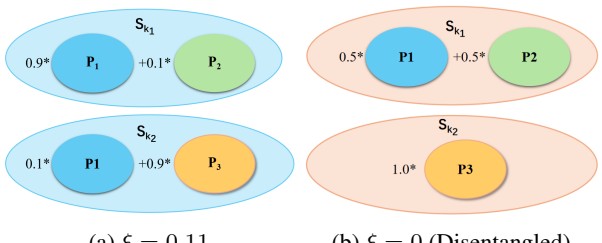

(a) $\xi = 0.11$      (b) $\xi = 0$ (Disentangled)

Figure 2: Two case of $\xi$-entangled: $\xi > 0$ (left) and $\xi = 0$ (right).

the data with a single label and domain), $\pi_i$ is the weight or probability assigned to the $i$-th base distribution, such that $\sum_{i=1}^{m} \pi_i = 1$ and $\pi_i > 0$. In the case of Gaussian Mixture Models (GMMs) (Reynolds et al., 2009), $P_i$ is the Gaussian distribution $\mathcal{N}(X | \mu_i, \Sigma_i)$. For each client's dataset $\mathcal{D}_k$ following a distribution $S_k$ can be represented as:

$$S_k = \sum_{i=1}^{m} \pi_{i,k} P_i, \tag{6}$$

where $\pi_{i,k}$ is the weight or probability assigned to the $i$-th base distribution, such that $\sum_{i \in \mathcal{C}_k} \pi_{i,k} = 1$ and $\pi_{i,k} \geq 0$. Noted that $\pi_{i,k} = 0$ if client $k$ doesn't have the distribution $S_i$. We define the Entangled coefficient between two clients based on the probability $\pi_{i,k}$:

**Definition 1.** *Let $\vec{\pi}_k = (\pi_{1,k}, \cdots, \pi_{m,k})$ denote the probability vector for client $k$. The Entangled coefficient $s_{k_1,k_2}$ between client $k_1$ and $k_2$ is defined as:*

$$s_{k_1,k_2} = \frac{< \pi_{k_1}, \pi_{k_2} >}{\|\pi_{k_1}\|_2 \|\pi_{k_2}\|_2} \tag{7}$$

*where $< a, b >$ represents the inner product of two vectors and $\| \cdot \|_2$ is the $\ell_2$ norm.*

Definition 1 quantifies the overlap in data distribution between clients by identifying the difference between probability on the base distribution. Obviously, $0 \leq s_{k_1,k_2} \leq 1$. We define two distributions to be orthogonal if $s_{k_1,k_2} = 0$.

According to the Entangled coefficient, we define $\xi$-entangled across clients' data distribution in federated learning according to the entangled coefficient.

---

[1]For various distribution estimation methods, refer to Section 2.2

**Definition 2.** *We define the distributions across $K$ clients as $\xi$-**entangled** if the Entangled coefficient $s_{k_1,k_2} \leq \xi$ for any two distinct clients $k_1, k_2 \in [K]$. Moreover, when $\xi = 0$, we define the distribution across $K$ clients to be **disentangled**.*

It is important to note that Entangled coefficient is closely related to the degree of Non-IID data in federated learning (FL) (Li et al.). Specifically, under IID conditions, where each client follows the same independent and identical distribution, this implies that $\pi_{i,k_1} = \pi_{i,k_2} > 0$ for any $k_1, k_2$ in Eq. (6). Consequently, $\xi$ equals 1. In contrast, under extreme Non-IID conditions, where each client has entirely distinct data distributions (e.g., client $k_1$ possesses a dataset consisting solely of 'cat' images, whereas client $k_2$ possesses a dataset exclusively of 'dog' images), the Entanglement Coefficient, $\xi$, equates to zero.

### 3.2.2 ANALYSIS ON DISENTANGLED AND NEAR-DISENTANGLED CASE

Numerous studies, as referenced in the literature (Kairouz et al., 2021; Li et al.), have demonstrated that the performance of the Federated Averaging (FedAvg) algorithm (McMahan et al., 2017) is significantly influenced under disentangled conditions. It will be demonstrated herein that there exists an algorithm capable of preserving model utility and efficiency within the disentangled scenario, with the distinct advantage of necessitating merely a single communication round.

**Theorem 1.** *If $f$ is $L$-lipschitz, a data distribution across clients being disentangled is a sufficient condition for the existence of a privacy-preserving federated algorithm that requires only a single communication round and achieves a utility loss of less than $\epsilon$ with a probability of at least $1 - 2\exp(-2\min\{n_1, \cdots, n_K\}\epsilon^2)$, i.e.,*

$$Pr(\epsilon_u \leq \epsilon) \geq 1 - 2\exp(-2\min\{n_1, \cdots, n_K\}\epsilon^2) \qquad (8)$$

Theorem 1 demonstrates that if the data distribution across clients is disentangled, it is possible to achieve an expected loss of $\epsilon$ with a probability of $1 - 2\exp\left(-2\min\{n_1, \ldots, n_K\}\epsilon^2\right)$ with only a single communication round. It is noteworthy that as the number of datasets $n_k$ or the expected loss tends to infinity, this probability approaches one. The implication of Theorem 1 is that when the distributions of individual clients are disentangled, clients can effectively transfer their distributions to the server. Subsequently, the server can aggregate the disentangled distributions uploaded by the clients in a single step to obtain the overall dataset distribution $\mathcal{D}$. Additionally, we provide an analysis of the small $\xi$ condition, referred to as near-disentangled.

**Theorem 2.** *When the distributions of across $K$ clients satisfy near-disentangled condition, specifically, $\xi < \frac{1}{(K-1)^2}$, if $f$ is $L$-lipschitz, then there exists a privacy-preserving federated algorithm that requires only one communication round and achieves a utility loss of less than $\epsilon$ with a probability of at least $1 - 2\exp(-\frac{(1-(K-1)\sqrt{\xi})n\epsilon^2}{2mL^2})$, i.e.,*

$$Pr(\epsilon_u \leq \epsilon) \geq 1 - 2\exp(-\frac{(1-(K-1)\sqrt{\xi})n\epsilon^2}{2mL^2}) \qquad (9)$$

Theorem 2 demonstrates that, within the framework of the near-disentangled scenario, the probability of achieving an expected loss of $\epsilon$ is given by $1 - 2\exp(-\frac{(1-(K-1)\sqrt{\xi})n\epsilon^2}{2mL^2})$. Specifically, as the entanglement coefficient $\xi$ increases, the probability decreases. For instance, in the disentangled case where $\xi = 0$, the probability equals one (see proof in Appendix).

## 4 THE PROPOSED ALGORITHM

This section introduces the proposed algorithm, called FedDistr, achieving one communication round while maintaining the model utility by leveraging the Latent Diffusion Model (LDM) Ho & Salimans (2022) due to its fast inference speed and the wide availability of open-source model parameters. FedDistr is mainly divided into the following three steps (see Fig. 3 and Algo. 1):

### 4.1 DISTRIBUTION DISENTANGLING

The first step is to disentangle the local data distribution into several base distribution for each client by leveraging the autoencoder part of the LDM. The autoencoder Van Den Oord et al. (2017);

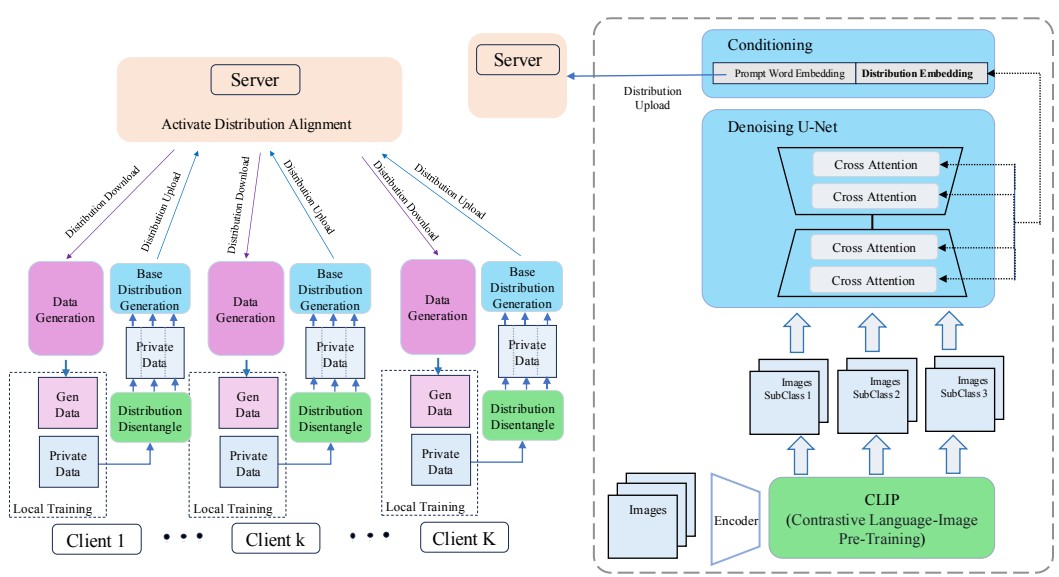

Figure 3: Overview of the proposed algorithm FedDistr.

Agustsson et al. (2017) consists of an encoder, $E$, and a decoder, $\mathcal{D}$. The encoder, $E$, maps input images $x$ to a spatial latent code, denoted as $z = E(x)$, while the decoder $\mathcal{D}$ is trained to perform the inverse mapping of the encoder, achieving an approximation $x \approx \mathcal{D}(E(x))$.

Specifically, each client $k$ initially encodes the private data into the latent feature embeddings as:

$$z_{k,i} = E(x_{k,i}), x_{k,i} \in \mathcal{D}_k \tag{10}$$

To disentangle the data distribution for client $k$, the data is segregated based on these latent feature embeddings. Furthermore, *client $k$ performs clustering* on the set $z_{k,i}{}_{i=1}^{n_k}$ to derive $m_k$ clusters by optimizing the objective function:

$$\min \sum_{z_{k,i} \in \mathcal{C}_j} \sum_{j=1}^{m_k} \|z_{k,i} - \bar{e}_j\|, \tag{11}$$

where $\mathcal{C}_j$ represents the set of points assigned to cluster $j$ and $\bar{e}_j$ is the centroid (mean) of cluster $j$. Thus, $\{z_{k,i}\}_{i=1}^{n_k}$ is separated into $m_k$ datasets as $\{B_{k,i}\}_{i=1}^{m_k}$. According to the separation result based on the clustering, we learn the disentangled distribution in the next section for each separated dataset.

## 4.2 DISTRIBUTION GENERATION

To estimate the base distribution $P$, each client learns their distribution parameters $v(P)$ by leveraging the embedding inversion technique in the diffusion model of LDM Ho & Salimans (2022); Liang et al. (2024). Specifically, a fixed prompt, such as "a photo of", is encoded using a frozen text encoder into a word embedding $p$. A learnable distribution parameter (prompt embedding) $v$ is then randomly initialized and concatenated with $p$ to form the guiding condition $[p; v]$. This combined condition is employed to compute the LDM's loss function. Our optimization goal to learn the distribution parameter $v_{k,i}$ of $i_{th}$ base distribution for client $k$ is formulated as:

$$v_{k,i} = \arg\min_v \mathbb{E}_{z \sim E(B_{k,i}), p, \epsilon \sim \mathcal{N}(0,1), t} [\|\epsilon - $$
$$\epsilon_\theta(\sqrt{\alpha_t} z + \sqrt{1 - \alpha_t}\epsilon, t, [p; v])\|_2^2], \tag{12}$$

where $z$ is the latent code of the input image $B_{k,i}$ generated by the encoder $E$, $\epsilon$ refers to noise sampled from the standard normal distribution $\mathcal{N}(0, 1)$, $t$ denotes the timestep in the diffusion process, $\alpha_t$ is a hyperparameter related to $t$ and $\epsilon_\theta$ is the model that predicts the noise. Therefore, we

derive the $m_k$ data representations $v_{k,i}$ by solving the optimization problem in Eq. (12). These representations are then fed into the diffusion model to generate the corresponding datasets for each disentangled base distribution $i$. In the federated learning setting, each client uploads their respective set of representations $v_{k,i}{}_{i=1}^{m_k}$ and corresponding data number to the server.

### 4.3 ACTIVE DISTRIBUTION ALIGNMENT

After each client $k$ uploads the distribution parameters $\{v_{k,i}\}_{i=1}^{m_k}$, the server first distinguishes between orthogonal and parallel data base distributions for different clients based on these parameters. Specifically, when the distance between distribution parameters, such as $\|v_{k_1,i} - v_{k_2,j}\|_2^2$, is small, it indicates that clients $k_1$ and $k_2$ share the same data base distribution.

Moreover, since the server does not have access to the sequence of the distribution parameters, matching these parameters across different clients presents a challenge. This matching problem can be framed as an assignment problem in a bipartite graph Dulmage & Mendelsohn (1958). To address this assignment problem, we utilize the Kuhn-Munkres algorithm (KM) Zhu et al. (2011), which is designed to find the maximum-weight matching in a weighted bipartite graph, or equivalently, to minimize the assignment cost. Specifically, the goal is to optimize the following expression for matching prompt embeddings between client 1 and $k$:

$$\min \sum_{i_1=1}^{m_{k_1}} \sum_{i_2=1}^{m_{k_2}} \|v_{k_1,i_1} - v_{k_2,i_2}\| e_{i_1,i_2}^{k_1,k_2} \qquad (13)$$

where $e_{i_1,i_2}^{k_1,k_2}$ represents the binary assignment variable, which equals 1 if cluster $i_1$ is assigned to cluster $i_2$, and 0 otherwise. Therefore, the server obtains multiple parallel sets $\mathcal{I}_p = \{(k,i)|e_{i_1,i_2}^{k_1,k_2} = 1, k \in [K], i \in [m_k]\}$. The orthogonal sets $\mathcal{I}_o = \{(k,i)|k \in [K], i \in [m_k]\} - \mathcal{I}_p$ consist of other pairs that are not part of the parallel sets.

Next, for the orthogonal sets, the server simply concatenates the parameters as follows $v_o = \bigcup_{k \in \mathcal{I}_o} v_{k,i}$.

For the parallel sets, the server selects one of the dominant distribution parameters (trained with the data number is the maximum) to represent all parameters in that set, which only requires a single communication round.

---

**Algorithm 1** FedDistr

**Input:** Local training epochs $T_l$, # of clients $K$, learning rate $\eta$, the dataset $\mathcal{D}_k = \{x_{k,i}, y_{k,i}\}_{i=1}^{n_k}$ owned by client $k$, pretrained latent diffusion model including a encoder $E$ and a generator $G$.
**Output:** $v$

1: ▷ *Clients perform:*
2: **for** Client $k$ in $\{1, \ldots, K\}$ **do**:
3:      Sample $x_i$ from $\mathcal{D}_k$;
4:      $z_{k,i} = E(x_{k,i})$
5:      Clustering $\mathcal{D}_k$ into $m_k$ datasets $\{B_{k,i}\}_{i=1}^{m_k}$ according to $z_{k,i}$ as Eq. (11).
6:      Initialize $\{v_{k,i}\}_{i=1}^{m_k}$.
7:      **for** $i$ in $[m_k]$ **do**:
8:          Learn the base distribution parameter $v_{k,i}$ by optimizing Eq. (12)
9:      Upload $\{v_{k,i}\}_{i=1}^{m_k}$ to the server;
10: ▷ *The server performs:*
11: Build the binary assignment $e_{i,j}^{k_1,k_2}$ via Eq. (13)
12: Obtain multiple parallel set $\mathcal{I}_p = \{(k,i)|e_{i_1,i_2}^{k_1,k_2} = 1, k \in [K], i \in [m_k]\}$
13: Obtain the orthogonal set $\mathcal{I}_o = \{(k,i)|k \in [K], i \in [m_k]\} - \mathcal{I}_p$
14: Obtain $v_o = \bigcup_{(k,i) \in \mathcal{I}_o} v_{k,i}$
15: Select the distribution parameter $v_p$ in $\mathcal{I}_p$ with the maximum trained data number;
16: Distribute $v = [v_p, v_o]$ to all clients;
17: **return** $v$

---

Finally, the server distributes the consolidated distribution parameters $v = [v_p, v_o]$ to all clients.

## 5 EXPERIMENT RESULTS

In this section, We present empirical studies to compare FedDistr with existing methods on utility, privacy and communication efficiency. Due to the page limit, please see the ablation study on more clients and large $\xi$ in Appendix B.

### 5.1 EXPERIMENTAL SETTINGS

**Models & Datasets.** We conduct experiments on two datasets: x'*CIFAR100* has the 20 superclass and each superclass has 5 subclass (Krizhevsky et al., 2014), thus, total 100 subclass; *DomainNet*

(Wu et al., 2015) has the 345 superclass (label) and each superclass has 5 subclass (style), thus, total 1725 subclass. We adopt ResNet (LeCun et al., 1998) for conducting the classification task to distinguish the superclass[2] on CIFAR100 and DomainNet. Please see details in Appendix A.

**Federated Learning Settings.** We simulate a horizontal federated learning system with K = 5, 10, 20 clients in a stand-alone machine with 8 Tesla V100-SXM2 32 GB GPUs and 72 cores of Intel(R) Xeon(R) Gold 61xx CPUs. For DomainNet and CIFAR10, we regard each subclass follow one sub-distribution ($P_i$). For the disentangled extent, we choose the averaged entangled coefficient $s = \frac{2}{K(K-1)} \sum_{k_1,k_2} d_{k_1,k_2}$ over all clients as 0, .... The detailed experimental hyper-parameters are listed in Appendix A.

**Baseline Methods.** *Four* existing methods i.e., FedAvg (McMahan et al., 2017): FedProx (Li et al., 2020b), SCAFFOLD (Karimireddy et al., 2020), MOON (Li et al., 2021) and the proposed method *FedDistr* are compared in terms of following metrics.

**Evaluation metric.** We use the (1 - model accuracy) to represent the utility loss, the rounds of required communication and the number of transmitted parameters to represent communication consumption, and the privacy budget in (Abadi et al., 2016) to represent the privacy leakage.

## 5.2 COMPARISON WITH OTHER METHODS

We first evaluate the tradeoff between utility loss and communication rounds in Fig. 4. Our analysis leads to two conclusions: 1) The proposed FedDistr achieves the best tradeoff between utility loss and communication rounds compared to other methods across various entangled cases ($\xi = 0, 0.003, 0.057$). For instance, in the disentangled case, FedDistr achieves a 40% utility loss with only one communication round, whereas MOON incurs approximately a 60% utility loss while requiring 100 communication rounds on CIFAR-100. 2) As the entanglement coefficient increases, other methods, such as FedProx, demonstrate improved performance, while FedDistr remains stable, consistently achieving a 40% utility loss on CIFAR-100 across different entangled scenarios.

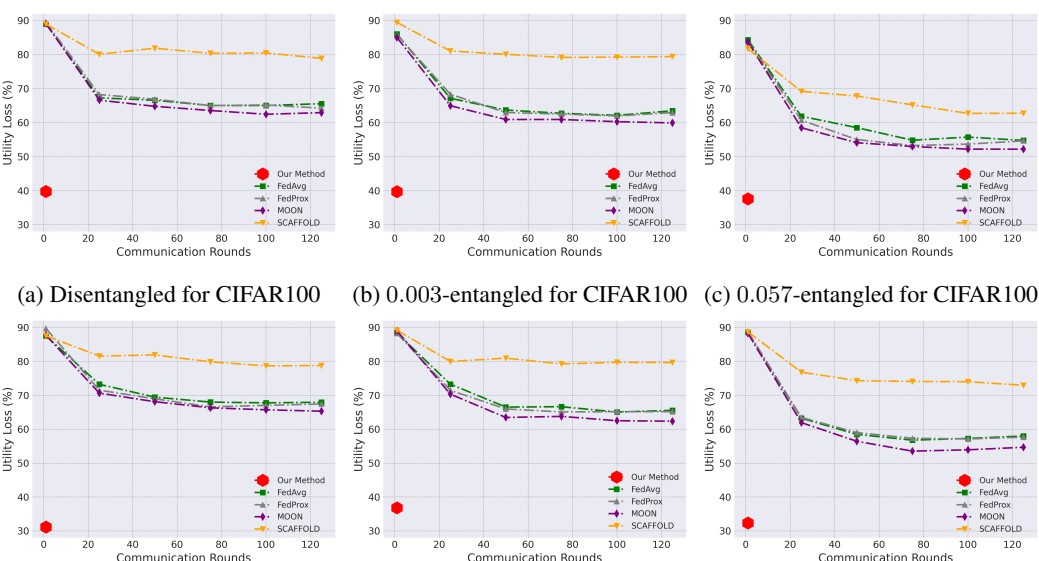

(a) Disentangled for CIFAR100    (b) 0.003-entangled for CIFAR100    (c) 0.057-entangled for CIFAR100

(d) Disentangled on DomainNet    (e) 0.003-entangled for DomainNet    (f) 0.057-entangled for DomainNet

Figure 4: Tradeoff between utility loss and communication round for different methods under different $\xi$-entangled scenario on CIFAR100 and DomainNet.

Moreover, we add random noises (Abadi et al., 2016) to protect transmitted parameters to evaluate the tradeoff between privacy leakage and utility loss in Fig. 5 . From this analysis, we can draw the following two conclusions: 1) The proposed FedDistr achieves the best tradeoff between utility loss

---

[2]each client only know the label of the superclass but doesn't know the label of subclass

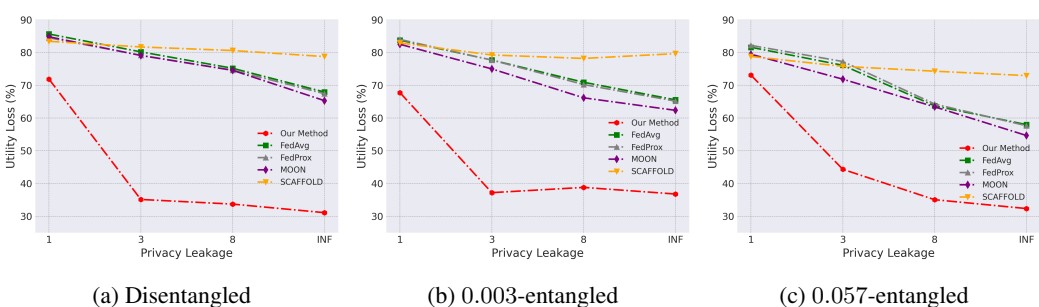

| (a) Disentangled | (b) 0.003-entangled | (c) 0.057-entangled |

Figure 5: Tradeoff between utility loss and privacy leakage for different methods under different $\xi$-entangled scenario on CIFAR100.

and privacy leakage; for instance, FedDistr (red line) is positioned in the lower-left region compared to other methods on both DomainNet and CIFAR-100. 2) As the noise level (privacy leakage) increases, the utility loss of different methods also increases.

Finally, we evaluate communication consumption, encompassing both the number of communication rounds and the amount of transmitted parameters, for five existing methods alongside our proposed method under disentangled and various $\xi$-entangled scenarios. The results summarized in Table 2 lead us to the following two conclusions: 1) Under near-disentangled or disentangled settings, the communication rounds required by FedDistr is only one, while FedAvg necessitates 80 rounds; 2) The number of transmitted parameters for the proposed FedDistr is merely 30K, in contrast to the 11.7M required by FedAvg for model transmission.

Table 2: Comparison on communication consumption until convergence (including communication rounds and number of the transmission parameters) for different methods under different $\xi$-entangled scenario.

| $\xi$ | Method | Communication rounds | Transmission parameters |
|---|---|---|---|
| 0 (Disentangled) | FedAvg | 150 | 11.7M |
| | FedDistr | 1 | 30K |
| 0.003 | FedAvg | 100 | 11.7M |
| | FedDistr | 1 | 30K |
| 0.057 | FedAvg | 80 | 11.7M |
| | FedDistr | 1 | 30K |

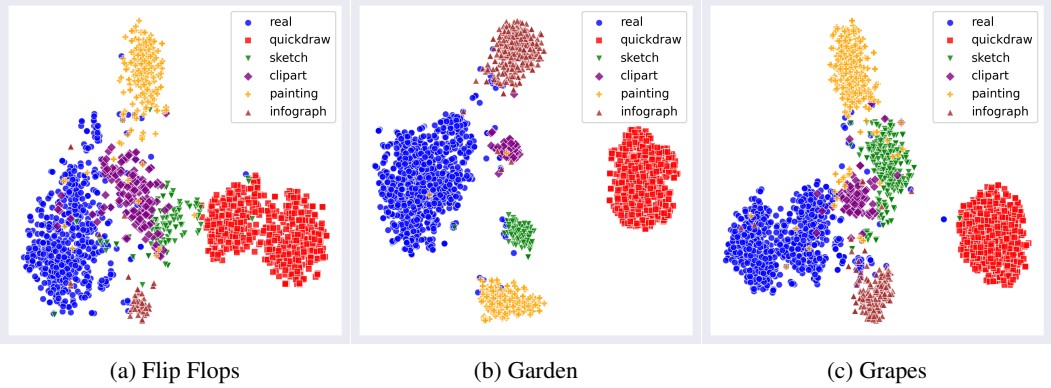

| (a) Flip Flops | (b) Garden | (c) Grapes |

Figure 6: Distribution Disentangling Visualization: clustering the encoded embedding $z_{k,i}$ illustrated in Sect. 4.1 of six subclass for each three superclass: Flip Flops, Garden, an Grapes

## 5.3 Visualization for distribution Disentangling

We also present the distribution disentangling result (clustering $z_{k,i}$ illustrated in Sect. 4.1) in Fig. 6. It shows that clustering on encoded embedding can separate the data with different base distribution well. Therefore, our disentangling method has a good effectiveness.

## 6 Discussion and Conclusion

In this study, we have addressed the critical inefficiencies inherent in Federated Learning (FL) due to the entanglement of client data distributions. By analyzing the distribution entanglement, we demonstrated that achieving a disentangled data structure significantly enhances both model utility and communication efficiency. Our theoretical analysis shows that under near-disentangled conditions, FL can achieve optimal performance with a single communication round, thus aligning closely with the efficiency of traditional distributed systems.

Furthermore, we propose the FedDistr algorithm by leveraging the diffusion model. The integration of stable diffusion models for data decoupling proves to be a robust solution, paving the way for future explorations in privacy-preserving machine learning. Ultimately, this work contributes to the advancement of FL by providing a clear pathway toward more efficient and effective federated learning.

With advancements in computational capabilities during the era of large language models (LLMs) (), the time consumption for local training has significantly decreased. Consequently, our focus is on enhancing communication efficiency. Moreover, the communication efficiency is especially important particularly in Wide Area Network scenarios (McMahan et al., 2017; Palmieri & Pardi, 2010).

Whether transferring data distribution leaks privacy is also an intriguing problem. Some studies (Xiao & Devadas, 2023) regard the underlying data distribution as a non-sensitive attribute, assuming that sharing insights into the distribution of data across users does not compromise individual privacy. However, other work indicates that using generative models to estimate the distribution still poses a risk of privacy leakage (Wu et al., 2021).

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

# A EXPERIMENTAL SETTING

This section provides detailed information on our experimental settings.

## A.1 DATASET & MODEL ARCHITECTURES

**Models & Datasets.** We conduct experiments on two datasets: *CIFAR100* Krizhevsky et al. (2014) and *DomainNet* Wu et al. (2015). For *CIFAR100*, we select 10 out of 20 superclasses, and 2 out of 5 subclasses in each superclass. For *DomainNet* Wu et al. (2015), we select 10 out of 345 superclasses (labels), and 3 out of 5 subclasses (domains) in each superclass. We adopt ResNet LeCun et al. (1998) for conducting the classification task to distinguish the superclass[3] on CIFAR100 and DomainNet.

**Federated Learning Settings.** We simulate a horizontal federated learning system with K = 5, 10, 20 clients in a stand-alone machine with 8 NVIDIA A100-SXM4 80 GB GPUs and 56 cores of dual Intel(R) Xeon(R) Gold 6348 CPUs. For DomainNet and CIFAR10, we regard each subclass follow one sub-distribution ($P_i$). For the disentangled extent, we choose the averaged entangled coefficient $s = \frac{2}{K(K-1)} \sum_{k_1,k_2} d_{k_1,k_2}$ over all clients as $0, ....$ The detailed experimental hyper-parameters are listed in Appendix A.

**Baseline Methods.** *Four* existing methods i.e., FedAvg McMahan et al. (2017): FedProx Li et al. (2020b), SCAFFOLD Karimireddy et al. (2020), MOON Li et al. (2021) and the proposed method *FedDistr* are compared in terms of following metrics.

**Evaluation metric.** We use the model accuracy on the main task to represent the utility, the communication rounds and transmission parameters to represent communication efficiency, and the standard derivation of noise level to represent the privacy leakage.

# B ABLATION STUDY

We evaluate the different methods for a large $\xi = 0.385$ in Figure 7. It shows that even for a large $\xi$, FedDistr achieves the best tradeoff between the communication rounds and utility loss. Moreover, compared to small $\xi$, other methods such as MOON converges to a better utility while it still requires beyond 100 communication rounds. Finally, we test the robustness of FedDistr for different number of clients in Figure 8. It illustrates FedDistr still achieve the better utlity than FedAvg with 20 clients.

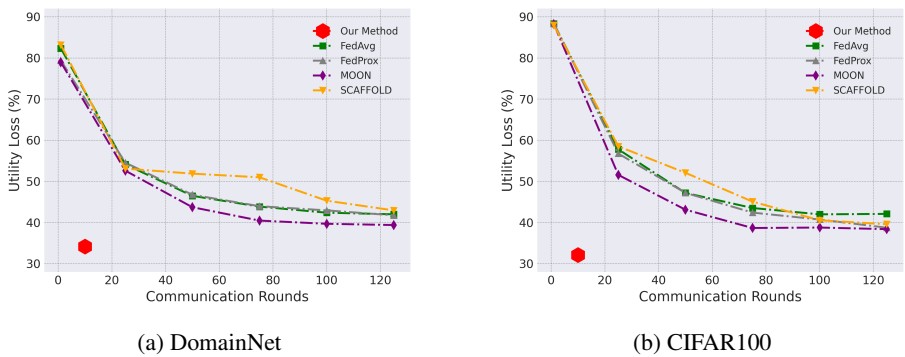

(a) DomainNet        (b) CIFAR100

Figure 7: Comparison on model accuracy for different methods under different $\xi$-entangled scenario.

---

[3]each client only know the label of the superclass but doesn't know the label of subclass

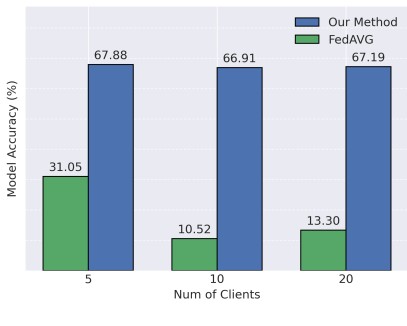

(a) DomainNet

Figure 8: Comparison on model accuracy for different number of clients

## C  THEORETICAL ANALYSIS

### C.1  PROOF FOR THEOREM 1

Consider $\mathcal{D} \sim S$ has total $n$ samples $\{X_1, \cdots, X_n\}$. We first demonstrate the estimated error of the distribution parameter on different data number $n$. In simplify for the analysis, we assume the $S$ is the Truncated normal distribution $\phi(\mu, \sigma, a, b)$ and only need to estimated parameter is $\mu$.

**Lemma 1.** *Hoeffding inequality. If $X_1, X_2, \ldots, X_n$ are independent random variables such that $X_i$ is bounded by [a, b] (i.e., $a \leq X_i \leq b$), and let $\bar{X} = \frac{1}{n} \sum_{i=1}^{n} X_i$ be the sample mean, then for any $\epsilon > 0$:*

$$P\left(|\bar{X} - \mu| \geq \epsilon\right) \leq 2 \exp\left(-\frac{2n\epsilon^2}{(a-b)^2}\right)$$

*where $\mu = E[X_i]$ is the expected value of the random variables.*

According to Hoeffding inequality, the probability of the estimated error for the distribution parameter $\mu$ is smaller than $2 \exp\left(-\frac{2n\epsilon^2}{(a-b)^2}\right)$. And this probability tends to zero if the $n$ tends to infinity.

We estimate the distribution mean $\mu$ via $\hat{\mu}$ according to the $\mathcal{D}$. Let the $\hat{\mu}$ be the mean of the estimated distribution $\hat{S}$. Define the loss function $f(\omega, z)$ on the model $\omega$ and data $z$. Define expectation loss function $F(\omega)$ and $F'(\omega)$ on data $S$ and $\hat{S}$ as:

$$F(\omega) = \mathbb{E}_{z \in S} f(\omega, z) \quad \text{and} \quad \hat{F}(\omega) = \mathbb{E}_{z \in \hat{S}} f(\omega, z) \tag{1}$$

**Lemma 2.** *Define $\omega^* = argmin_\omega F(\omega)$ and $\hat{\omega}^* = argmin_\omega \hat{F}(\omega)$. If $f$ is L-lipschitz, then the following holds with the probability $2 \exp\left(-2n\epsilon^2\right)$:*

$$0 \leq \mathbb{E}_{z \in S} f(\hat{\omega}^*, z) - \mathbb{E}_{z \in S} f(\omega^*, z) \leq 2L\epsilon \tag{2}$$

*Proof.* According to definition of $\omega^*$ and $\hat{\omega}^*$, we have

$$\mathbb{E}_{z \in \hat{S}} f(\hat{\omega}^*, z) \leq \mathbb{E}_{z \in \hat{S}} f(\omega^*, z) \tag{3}$$

and

$$\mathbb{E}_{z \in S} f(\omega^*, z) \leq \mathbb{E}_{z \in S} f(\hat{\omega}^*, z). \tag{4}$$

Therefore, we can obtain

$$\mathbb{E}_{z \in S} f(\hat{\omega}^*, z) - \mathbb{E}_{z \in S} f(\omega^*, z) \geq 0 \tag{5}$$

Furthermore, the following holds with the probability $1 - 2\exp\left(-2n\epsilon^2\right)$ based on Lemma 1:

$$
\begin{aligned}
&\mathbb{E}_{z \in S} f(\hat{\omega}^*, z) - \mathbb{E}_{z \in S} f(\omega^*, z) \\
&\leq [\mathbb{E}_{z \in S} f(\hat{\omega}^*, z) - \mathbb{E}_{z \in \hat{S}} f(\hat{\omega}^*, z)] + [\mathbb{E}_{z \in \hat{S}} f(\omega^*, z) - \mathbb{E}_{z \in S} f(\omega^*, z)] \\
&\leq \int_{z \in S} |f(\hat{\omega}^*, z_1) - f(\hat{\omega}^*, z_1 + \epsilon)| dp(z) + \int_{z \in S} |f(\omega^*, z_1) - f(\omega^*, z_1 + \epsilon)| dp(z) \quad (6) \\
&\leq L(z_1 + \epsilon - z_1) + L(z_1 + \epsilon - z_1) \\
&= 2L\epsilon,
\end{aligned}
$$

where $p(z)$ is the probability density function. The last inequality is due to the L-lipschitz of $f$. $\quad\square$

Lemma 2 demonstrates the utility loss when transferring the estimated distribution. Consider $K$ clients participating in federated learning with their data $\mathcal{D}_k = \{(x_{k,i}, y_{k,i})\}_{i=1}^{n_k}$. Denote $\mathcal{D} = \mathcal{D}_1 \cup \cdots \cup \mathcal{D}_K$ and $\mathcal{D}$ follows the distribution $S$. Then we prove Theorem 1 according to Lemma as follows:

**Theorem 1.** *A data distribution across clients being disentangled is a sufficient condition for the existence of a privacy-preserving federated algorithm that requires only a single communication round and achieves the $\epsilon$ utility loss with the probability $1 - 2\exp(-\frac{\min\{n_1,\cdots,n_K\}\epsilon^2}{2L^2})$.*

*Proof.* If the data distribution across clients is disentangled, it means each client $k$ can use their own $n_k$ data to estimate their data distribution. Therefore, according to Lemma 2, when achieving utility loss $\epsilon$ for distribution $S_k$ (i.e., $\mathbb{E}_{z \in S_k} f(\hat{\omega}^*, z) - \mathbb{E}_{z \in S_k} f(\omega^*, z) \leq \epsilon$), the probability is $2\exp(-\frac{n_k\}\epsilon^2}{2L^2(a-b)^2})$.

Then each clients can transfer estimated $\hat{S}_k$ to the server, thus,

$$
\begin{aligned}
&\mathbb{E}_{z \in S} f(\hat{\omega}^*, z) - \mathbb{E}_{z \in S} f(\omega^*, z) \\
&= \frac{1}{K} \sum_{k=1}^{K} \mathbb{E}_{z \in S_k} f(\hat{\omega}^*, z) - \mathbb{E}_{z \in S_k} f(\omega^*, z) \quad (7) \\
&\leq \frac{1}{K} K\epsilon = \epsilon.
\end{aligned}
$$

And the probability to achieve $\epsilon$ utility loss is $1 - 2\exp(-\frac{\min\{n_1,\cdots,n_K\}\epsilon^2}{2L^2})$.

$\square$

## C.2 ANALYSIS ON NEAR-DISENTANGLED CASE

**Lemma 3.** *Consider $K$ unit vectors $\vec{a}_k = (a_{1,k}, \cdots, a_{m,k})$ such that $\sum_{k=1}^{K} a_{i,k} = 1$ for any $i$, if $<\vec{a}_i, \vec{a}_j> \leq \xi$ for any $i \neq j$ and $\xi < \frac{1}{(K-1)^2}$, then $\max\{a_{i,1}, \cdots, a_{i,K}\} \geq 1 - (K-1)\sqrt{\xi}$*

*Proof.* Since $<\vec{a}_i, \vec{a}_j> \leq \xi$ for any $i \neq j$, $\min\{a_{i,k_1}, a_{i,k_2}\} \leq \sqrt{\xi}$ for any $i, k_1 \neq k_2$. Therefore, we have

$$
\begin{aligned}
\max\{a_{i,1}, \cdots, a_{i,K}\} &\geq 1 - \sum_{k \neq 1} \min\{a_{i,1}, a_{i,k}\} \\
&\geq 1 - (K-1) \min\{a_{i,1}, a_{i,k}\} \quad (8) \\
&\geq 1 - (K-1)\sqrt{\xi}
\end{aligned}
$$

$\square$

For the near-disentangle case (that small $\xi$), we have the following theorem:

**Theorem 2.** *When the distributions of across $K$ clients satisfy near-disentangled condition, specifically, $\xi < \frac{1}{(K-1)^2}$, then there exists a privacy-preserving federated algorithm that requires only one communication rounds and achieves the $\epsilon$ expected loss error with the probability $2\exp(-\frac{(1-(K-1)\sqrt{\xi})n\epsilon^2}{2mL^2})$.*

*Proof.* In simplify the analysis, we assume $\vec{\pi}_k$ to be unit vector and the number of data for each base distribution to be the same ($\frac{m}{n}$), i.e., $\sum_{k=1}^{K} \pi_{i,k} = 1$. According to the Lemma 3, we have

$$\max\{a_{i,1}, \cdots, a_{i,K}\} \geq (1 - (K-1)\sqrt{\xi}) \tag{9}$$

which means for each base distribution, there exists one client has at least the $\frac{(1-(K-1)\sqrt{\xi})m}{n}$ data. Thus, the probability when achieving the utility loss less than $\epsilon$ for any base distribution $P_i, 1 \leq i \leq m$ is

$$\exp(-\frac{(1-(K-1)\sqrt{\xi})n\epsilon^2}{2mL^2})$$

according to the lemma 2.

Therefore, the probability when achieving the utility loss on $S$ less than $\epsilon$ is

$$\min_{k \in [K]} \{2\exp(-\frac{(1-(K-1)\sqrt{\xi})n\epsilon^2}{2mL^2})\} = 2\exp(-\frac{(1-(K-1)\sqrt{\xi})n\epsilon^2}{2mL^2}).$$

$\square$

