# OpenReview forum: "Disentangling data distribution for Federated Learning"
_ICLR.cc/2025/Conference — ICLR 2025 Conference Withdrawn Submission_

### Official Review · Reviewer_nZ5k · 2024-10-17

**Soundness:** 2
**Presentation:** 2
**Contribution:** 2
**Rating:** 6
**Confidence:** 5

**Summary:**

The paper proposes an one-shot method approach to federated learning by introducing the concept of “disentangling” distributions. The authors determine two distributions to be disentangled if given the same mixed distributions, the stochastic weight vector has a cosine similarity of 0. The authors theoretically support their approach via Theorems 1 and 2. The proposed algorithm, FedDistr, trains a Latent Diffusion Model which learns to generate distributions which are aligned in a central server. The alignment process uses the distribution parameters to determine orthogonal and parallel distributions for different clients; this is accomplished by solving an optimization problem on a weighted bipartite graph. The consolidated distribution parameters are broadcasted to all clients.

The work provides a good, singularly novel, idea, that of entangled/disentangled distributions and the creation of parallel/orthogonal sets of parameters. The body of the work relies heavily on the work of Liang et al. (2024), "Diffusion-Driven Data Replay: A Novel Approach to Combat Forgetting in Federated Class Continual Learning." Unlike in Liang et al., which thoroughly explains each step of their algorithm, the authors sweep factors underneath the rug. For example, Liang et al. state that the Latent Diffusion Model (LDM) is used to generate images at the edge which are used to train a classifier. The classifier, trained locally, is aggregated through the use of the traditional FedAvg algorithm. The authors of this paper do not clarify if they do or do not do this. It must be assumed by the reader that they do not, but one who reads the related work I have referenced will wonder if that is truly the case. In fact, I believe a re-organization and further clarifications would strengthen the paper significantly.  The authors did not reference Liang et al. and break down their work in the Related Works section; this is an obvious mistake as the two are without a doubt built on the same core with a minor, although interesting and significant, improvement. I believe it would be beneficial to the paper if the authors added a flow chart or table comparing/contrasting the work of Liang et al. and their work.

If the authors had more clearly explained how a final classifier is made, if one was even trained, I would be willing to increase the score. It would be prudent to add a subsection on how classification works, if a model is trained or not, and how one achieves a singular classification model used by all agents. I believe the paper has good merit but with further clarifications and much clear writing and flow, the paper would be much stronger than it currently is. A key problem with the writing is the authors are relying on too much prior knowledge rather than opting in for clarity and verbosity, both of which would make it a much better read with much clearer communication of ideas. Furthermore, I believe the experiments section needs to be clearer with regard to what was done. The additional, but few, comments in the Appendix mention the authors used subsets of the classes of the datasets; what does this mean and how is the data distributed to the agents? If 20 classes are used and 5 clients exist in your network, are you forcing each agent to have a different subset of the 20 classes to force heterogeneity? This further speaks to the overall lack of clarity in the writing and explanation provided by the authors. I would ask the authors to better explain their experimental setup and the associated variables. I would also ask the following: How many classes of each dataset were used in total? How were these classes distributed among the clients? Was heterogeneity artificially induced, and if so, how?

Overall, independently of the lack of clarity, I believe the paper is relevant to readers in the field of Federated Learning, as it proposes an interesting one-shot approach to heterogeneous federated learning. If possible, I would prefer to see the paper rewritten for clarity but the results and ideas are important enough to warrant acceptance.

**Strengths:**

The primary, and possibly only, strength of the paper is the ability to achieve high accuracy scores with only one communication round in a heterogeneous federated learning scenario; this, followed by theoretical support proved in the appendix, the authors proposed a unique solution to a commonly known problem in federated learning. The idea of disentangling the local data is original and significant.

**Weaknesses:**

The weaknesses of the paper are all indirectly related to their contribution. The writing lacks extreme amounts of clarity all around. The authors are not clearly explaining concepts and do not mention how extremely similar the core of their work is to Liang et al. (2024). They cited the paper in a small section of the paper, but it is much more significant than they lead the readers to believe and because of this it needs to be introduced in the related works section with a proper break down. The authors further fail to clarify if the ResNet classifier is trained and fused or if the one-communication step is only to learn to generate images and labels through the Latent Diffusion Model. These are important questions. Furthermore, there are several things to cover that I will double-up and put in the "Questions" section because I believe the authors need to answer them. Here they are:
1) In the experiments section, the authors did not explain the exact precisely enough what the federated architecture looks like. While stating that K=5,10,20, does this mean there are a total of K clients in the network and all K are used when training? It is typical to show results with a subset of the agents as the contributing agents in the learning step. For example, in the work that introduced FedAvg, the authors design a network with 100 agents and use subsets as the contributing agents when fusing models, such as 10 of the 100 are used to average their models together.
2) The authors also don’t clearly state what the distribution of the data looks like at the edge. While their proposed algorithm aim to disentangle the distributions, one wonders what the underlying initial distribution of data does to the result of the model. Perhaps the authors’ goal was to limit to cases of disentangled and nearly-disentangled, but it would be better to make this clear. Furthermore, of the subclasses of the datasets selected, how are those samples distributed? This is also not clearly defined.
3) The paragraph between eq 2 and 3 contains the unsubstantiated claim, other than their own claim on the matter via the presented “Theorem 1”: “Both theoretical analysis and empirical studies show that data heterogeneity is a blessing rather than a curse, as long as data distributions among different clients can be completely disentangled.” This claim needs to be supported by citations such that readers can confirm such a bold claim, which is commonly known to be a difficult hurdle for FL-algorithms to overcome.
4) In the paragraph between equation 10 and 11, the authors state the following: “To disentangle the data distribution for client k, the data is segregated based on these latent feature embeddings.” What exactly does this mean and how is this accomplished?
5) The second paragraph of section 4.3, “Active Distribution Alignment,” needs clarification on what it meant by “the server does not have access to the sequence of the distribution parameters.” Further, when the authors state “orthogonal and parallel data base distributions,” what does this mean?
6) The proofs in the appendix need more clarifications to be more clear and need grammatical cleaning (e.g., the proof of Theorem 1 has a "}" in the "exp( . )" above equation 7).

**Questions:**

I have a singular, but deep question that I need answered, because it is not at all clarified in the paper: Could you please clarify how a singular, global, classification model is obtained? In the referenced work of Liang et al., a clearly important reference as the authors have a very similar structure to the one presented, the authors learn to generate datasets and labels via the LDM, but further train a classifier in a FedAvg fashion. It seems to me that this work fixes the multiple-communication rounds used to learn the distribution parameters used to generate samples/labels at the edge, but speaks nothing about the classifier model itself.

The answer to this question is at the heart of the paper itself and needs to be answered. If a classifier is trained, how is it done? How does it perform on a test set that includes labels outside of the local training distribution? Does it require aggregation, as is typically done in federated learning?

---

> ### Author Response · Authors · 2024-12-04
>
> **Q1. In the experiments section, the authors did not explain the exact precisely enough what the federated architecture looks like.**
>
> We are sorry for this unclear claim.  Our experimental setting is all $K$ clients participate for each communication round. We would include more clients sampling strategy in the future version.
>
> **Q2. The authors also don’t clearly state what the distribution of the data looks like at the edge.**
>
> The data is distributed according to the $\epsilon$. The small $\epsilon$ indicates the large heterogeneity of the data distribution among clients.
>
> **Q3. In the paragraph between equation 10 and 11, the authors state the following: “To disentangle the data distribution for client k, the data is segregated based on these latent feature embeddings.” What exactly does this mean and how is this accomplished?**
>
> We apologize for the unclear clarification. This statement means that each client applies a clustering algorithm to the latent feature embeddings to disentangle their own complex distributions. The client then uploads some of these disentangled base distributions to the server.
>
> **Q4.  The second paragraph of section 4.3, “Active Distribution Alignment,” needs clarification on what it meant by “the server does not have access to the sequence of the distribution parameters”.**
>
> The server receives the distribution parameters uploaded by different clients. Then it aligns these distributions to identify the orthogonal ($\mathcal{I}_o$) and parallel sets ($\mathcal{I}_p$).
>
> **Q5. The authors state “orthogonal and parallel data base distributions”,  what does this mean?**
>
> The orthogonal set refers to the set of prompts that are largely different, while the parallel set refers to the set of prompt embeddings where the prompts are similar.
>
> **Q6. Could you please clarify how a singular, global, classification model is obtained?**
>
> We apologize for the confusion. The detailed process is as follows.
>
> After the server actively aligns the uploaded base distributions, each client receives the distributed base distributions $v=[v_p,v_o]$ from server. For each base distribution embedding $v_o^i \in v_o$, clients use the latent diffusion model to generate data $\widehat{x} _ i=\{\widehat{x} _ {i,j} | j\in [n_i] \}$ following the base distribution.
> $$\widehat{x} _ {i,j} = D(f(z _ {T}, T, [p, v_o^i])), j \in [n_i] \text{, for } i \in [|v_0|],$$
> where decoder $D$ is for the inverse mapping of encoder $E$, $f$ is the denoising process, $T$ is the number of diffusion steps, and $[p, v_i]$ is the conditional prompt corresponding to base distribution $i$.
>
> Using the synthetic data, the clients locally train their downstream task model.
> $$\mathcal{F} _ {k}^{*} = \text{argmin} _ {\mathcal{F} _ {k}} \mathbf{E} _ {i\in [|v_0|]} \mathcal{L} _ {CE} (\mathcal{F} _ {k}(\widehat{x} _ {i,j}), \widehat{y} _ i) \text{, for } k \in [K],$$
> where $\mathcal{F} _ k$ is the local downstream model of client $k$, $\mathcal{L} _ {CE}$ is the cross entropy loss, $\{\widehat{x} _ {i,j} | j\in [n_i] \}$ is the generated data corresponding to base distribution $i$.
>
> **Q7. If a classifier is trained, how is it done? How does it perform on a test set that includes labels outside of the local training distribution?**
>
> We apologize for the confusion. First, the classifier is locally trained on synthetic data generated from the base distributions without any aggregation. Second, we evaluate the trained model using a test dataset that includes all the labels to assess whether achieving a global optimum [1,2].
>
> [1]Li T, Sahu A K, Zaheer M, et al. Federated optimization in heterogeneous networks[J]. Proceedings of Machine learning and systems, 2020, 2: 429-450.
>
> [2]Li Q, He B, Song D. Model-contrastive federated learning[C]//Proceedings of the IEEE/CVF conference on computer vision and pattern recognition. 2021: 10713-10722.

---

### Official Review · Reviewer_TSra · 2024-10-30

**Soundness:** 1
**Presentation:** 3
**Contribution:** 2
**Rating:** 3
**Confidence:** 4

**Summary:**

The paper tackles the problem of heterogeneous federated learning by estimating the data distribution in a federated way. Specifically, it assumes that the global data distribution can be represented as a mixture of a set of base distributions, with local distributions varying from the global one in their mixture coefficients. The proposed approach uses diffusion models to disentangle data into base distributions and communicates these distributions using only a single round of communication.

**Strengths:**

- Novel application of diffusion models to disentangle and aggregate data distributions in federated learning.
- Demonstrates some improvements in communication efficiency and utility for specific datasets.

**Weaknesses:**

- The approach is similar to previous approaches [1, 2] but does not discuss them.
- The theoretical results are questionable and assumptions are not clearly stated.
- The results seem to contradict previous theoretical results [1].
- The description of the approach omits important details, in particular, how the local models are trained and if local models can differ.

**Questions:**

- Marfoq et al. [1] provide an impossibility result that shows that federated learning under a mixture of distributions is impossible without further assumptions on the distributions. It seems that this result is applicable to your case. How can this be reconciled with your paper?
- Shamir and Srebro [3] show that one-shot distributed stochastic learning via averaging of parameters can be arbitrarily bad. How does this result relate to your work?
- It can be shown that one-shot federated learning is provably efficient for convex learning problems when using robust aggregation [4]. How does this relate to your approach?
- I have troubles understanding why Lemma 2 should be correct. It seems that without further assumptions on $S$ and $\widehat{S}$ we cannot bound the expected loss. Assume that under $S$ only $z$ has probability 1 and all others have probability 0 and that under $\widehat{S}$ some $z'$ has probability 1 and all others have probability 0. Now we can assume that $f(w,z)=0$ for all $w$ and for any number $C>2L\epsilon$, we can assume $f(w,z')=C$. Then the bound in Lemma 2 should not work. Did I understand an assumption wrongly?
- The proof of Thm 1 heavily relies on Lemma 2. It basically says: if local distributions have nothing to do with each other, then we can estimate the global distribution from local distributions. Now with Lemma 2 this would imply that one can learn a good model on that. This not only seems to contradict the impossibility result in Marfoq et al., it also seems to contradict classical transfer learning results - if local tasks have nothing to do with each other, how can we learn from each other?
- The paper does not state in the main text that for the theoretical results to hold, one has to find the global risk minimizer (as stated in the assumptions of Lemma 2). Since this is in practice infeasible for non-convex problems, such as deep learning, the paper should address the limitations of their theory.
- How is the model trained? Is it trained on the estimate of the global distribution $\widehat{S}$ by generating examples and training on them? How can such a model be good if local distributions are disentangled?
- SCAFFOLD generally performs well on heterogeneous FL tasks; however, your results show it underperforms. Could you elaborate on potential reasons for this discrepancy and clarify whether experimental settings or parameters might have affected SCAFFOLD's performance?


References:

[1] Marfoq, Othmane, et al. "Federated multi-task learning under a mixture of distributions." Advances in Neural Information Processing Systems 34 (2021): 15434-15447.

[2] Wu, Yue, et al. "Personalized federated learning under mixture of distributions." International Conference on Machine Learning. PMLR, 2023.

[3] Shamir, Ohad, and Nathan Srebro. "Distributed stochastic optimization and learning." 2014 52nd Annual Allerton Conference on Communication, Control, and Computing (Allerton). IEEE, 2014.

[4] Kamp, Michael, et al. "Effective parallelisation for machine learning." Advances in Neural Information Processing Systems 30 (2017).

---

> ### Author Response · Authors · 2024-12-04
>
> **Q1. The approach is similar to previous approaches but does not discuss them.**
>
> **Regarding [1,2,3]:** Our paper focuses on improving communication efficiency  while other papers [1,2,3] concentrated on enhancing the model utility (global and personalized performance).
>
> **Regarding [4]**: While our paper primarily aims to reduce the communication rounds, [4] focuses on minimizing computational complexity.
>
> [1] Marfoq, Othmane, et al. "Federated multi-task learning under a mixture of distributions." Advances in Neural Information Processing Systems 34 (2021): 15434-15447.
>
> [2] Wu, Yue, et al. "Personalized federated learning under mixture of distributions." International Conference on Machine Learning. PMLR, 2023.
>
> [3] Shamir, Ohad, and Nathan Srebro. "Distributed stochastic optimization and learning." 2014 52nd Annual Allerton Conference on Communication, Control, and Computing (Allerton). IEEE, 2014.
>
> [4] Kamp, Michael, et al. "Effective parallelisation for machine learning." Advances in Neural Information Processing Systems 30 (2017).
>
> **Q2. The theoretical results are questionable and assumptions are not clearly stated.**
>
> I apologize for the earlier lack of clarity. To clarify, $S$ and $\hat{S}$ represent truncated Gaussian distributions, and the mean difference between them is bounded by $\epsilon$. Based on this, we derive Eq.(6).
>
> The intuition behind Theorem 1 is that when local distributions are independent, they can be concatenated into a single global distribution. This global distribution can generate data consistent with any of the local distributions. As a result, the generated data can effectively support the training of the downstream model.
>
> **Q3. The results seem to contradict previous theoretical results [1].**
> The conclusion is not in contradiction with [1]. The key difference lies in the aggregation approach: while [1] averages all model parameters, FedDistr adopts a more selective strategy by disentangling the parameters. Specifically, FedDistr actively separates the disentangled and entangled components, aggregating the entangled parts multiple times while concatenating the disentangled parts only once.
>
> [1] Marfoq, Othmane, et al. "Federated multi-task learning under a mixture of distributions." Advances in Neural Information Processing Systems 34 (2021): 15434-15447.
>
> **Q4. The description of the approach omits important details, in particular, how the local models are trained and if local models can differ.**
>
> Please refer to Q3 of Reviewer mZWU for further details.
>
> **Q5. SCAFFOLD generally performs well on heterogeneous FL tasks; however, your results show it underperforms.**
>
> This is because this paper mainly focus on the disentangled and near-disentangled scenario, i.e., the data distribution of clients are extreme Non-IID. Therefore, SCAFFOLD doesn't perform well.

---

### Official Review · Reviewer_n8vX · 2024-11-04

**Soundness:** 3
**Presentation:** 3
**Contribution:** 2
**Rating:** 5
**Confidence:** 4

**Summary:**

This paper proposes a novel FedDistr algorithm, which employs stable diffusion models to transfer data distributions instead of model parameters between clients and the server. The theoretical and experimental results demonstrate the effectiveness of the proposed method. Overall, the paper is easy to understand, but I still have some concerns regarding the fundamental settings in federated learning and the risks of privacy leakage.

**Strengths:**

1. The paper is easy to follow.
2. The experiment results look good.

**Weaknesses:**

1. The proposed method may have the potential to leak privacy.
2. Some related work on methods that transmit knowledge instead of models between clients and the server may need to be reviewed and discussed.
3. There are some minor spelling and grammatical errors.

**Questions:**

1. There are some grammatical errors.
   - “FL is hindered by entanglement of data distributions across different clients” —> “FL is hindered by **the** entanglement of data distributions across different clients”
   - “both disentangled base distribution” —>  “both disentangled base distributions”
   - “…for achieving ideal efficiency of federated learning…” –> “…for achieving **the** ideal efficiency of federated learning…”
   - ”extract data distributions via stable diffusion model, and then upload these decoupled distributions to the server” —>  ”extract data distributions via **a** stable diffusion model and then upload these decoupled distributions to the server”
   - “The server actively identifies the orthogonal or parallel between the base distributions uploaded by clients and aggregate the orthogonal distribution once.” —> “The server actively identifies the orthogonal or parallel between the base distributions uploaded by clients and **aggregates** the orthogonal distribution once.”

2. In the reference “Numerous studies (… Li et al..),” the year is missing; it’s recommended to update the references.

3. It is better to adjust the formatting of the references. For example, ...
   - Use**\citet**: “Latent Diffusion Model (LDM) Ho & Salimans (2022)” —>“Latent Diffusion Model (LDM, Ho & Salimans, 2022) ”
   - Use**\citep**“: The autoencoder Van Den Oord et al. (2017); Agustsson et al. (2017) “ —>  “The autoencoder (Van Den Oord et al., 2017; Agustsson et al., 2017) “
   - Use**\citep**“: LDM Ho & Salimans (2022); Liang et al. (2024)” –> “LDM (Ho & Salimans, 2022; Liang et al., 2024)”

4. Some equations lack punctuation at the end, such as Eqs. (8), (9), (10) …

5. Some minor spelling issues ：
   - “consisting solely of ’cat’ images, … ’dog’ images” —> “consisting solely of **\``cat‘’ **images, … **\``dog’’** images”.
   - “Theorem 2. When the distributions of across K clients satisfy” —> “Theorem 2. When the distributions across K clients satisfy”
   - “We conduct experiments on two datasets: x‘CIFAR100 has the 20…” —> “We conduct experiments on two datasets: CIFAR100 has the 20…”
   - “For DomainNet and CIFAR10, we regard…” —> “For DomainNet and CIFAR100, we …”
   - “the era of large language models (LLMs) (), the time” —> “the era of large language models (LLMs) , the time”

6. It is recommended to present Algorithm 1 in a single-line format to avoid confusing indentations and spacing.

7. In Section 4.3, each client $ k$ uploads the distribution parameters to the server, but the paper states that “the server does not have access to the sequence of the distribution parameters.” This seems contradictory.

8. The idea of transferring information contained in data rather than the data itself has been reflected in some previous works, such as methods based on distillation and those using data Mixup. It is recommended to include a review and discussion of similar methods in the paper, and ideally, to add these baselines in the experiments.

9. If clients and the server have information about the base distributions, can they infer the distribution information of other clients? Would this potentially lead to privacy leakage? It would be beneficial to include a discussion in the paper regarding the privacy and security implications of the proposed method.

**Details Of Ethics Concerns:**

N.A.

---

> ### Author Response · Authors · 2024-12-04
>
> **Q1. The proposed method may have the potential to leak privacy.**
>
> In this paper, clients transfer the base data distribution with DP between clients and server, which protect clients' data privacy. For example, we add random noise following normal distribution to the uploaded base distribution embeddings, aiming satisfy differential privacy. In Algorithm 1, each client upload $\{\widetilde{v} _ {k,i}\} _ {i=1}^{m_k}$ to the server.
> $$\widetilde{v} _ {k,i} = v _ {k,i} / max\{ \lbrace 1, \frac{\|v _ {k,i}\| _ 2}{C} \rbrace \} + \mathcal{N}(0, \sigma^2 C^2 I), i \in [m_k], \text{ for } k \in [K].$$where $C$ is the norm upper bound, $\sigma$ is the standard derivation of added noise.
>
> Moreover, we recognize the potential leakage of group statistics, such as data means. Techniques such as sparsification[1] and quantization[2] of the uploaded distribution parameters may help mitigate this risk. We plan to explore the protection of these aspects in future work.
>
> Finally, we emphasize that the small size of the transferred parameters and the single communication round make this approach compatible with homomorphic encryption (HE) methods. Using HE would allow computations on encrypted embeddings directly, further eliminating the risk of privacy leakage without significant additional computational overhead. Incorporating HE is an avenue we plan to investigate for scenarios requiring stronger privacy guarantees.
>
> [1]Aji A F, Heafield K. Sparse communication for distributed gradient descent[J]. arXiv preprint arXiv:1704.05021, 2017.
>
> [2]Shlezinger N, Chen M, Eldar Y C, et al. UVeQFed: Universal vector quantization for federated learning[J]. IEEE Transactions on Signal Processing, 2020, 69: 500-514.
>
> **Q2. Some related work on methods that transmit knowledge instead of models between clients and the server may need to be reviewed and discussed.**
>
> Thanks for your constructive suggestion. Prior works have explored the idea of transmitting knowledge about private data, such as methods based on knowledge distillation and data Mixup. However, these approaches still require lots of communication rounds.
>
> Specifically, knowledge distillation methods frequently require multiple communication rounds and may introduce significant communication overhead due to the transmission of auxiliary models or generators. For example, FedDF[3] distills the ensemble of client teacher models into a server-side student model, necessitating multiple exchanges of logits. Similarly, [4] proposes a data-free knowledge distillation approach where the server learns a generator to aggregate user information, requiring repeated transmission of a lightweight generator. Methods based on data Mixup, such as [5], also involve multiple communication rounds, as they rely on transmitting a batch of averaged data per round.
>
> In contrast, our proposed FedDistr significantly improve the communication efficiency, which only requires one round of transmitting embeddings. In FedDistr, clients more accurately extract the data distribution by using the latent diffusion model and upload data distributions represented by LDM conditional embeddings. The server actively aligns the conditional embeddings and distribute the embedding to the clients.
>
> We will include a discussion of these related works and add the methods as baseline in the revised manuscript.
>
> [3]Lin T, Kong L, Stich S U, et al. Ensemble distillation for robust model fusion in federated learning[J]. Advances in neural information processing systems, 2020, 33: 2351-2363.
>
> [4]Zhu Z, Hong J, Zhou J. Data-free knowledge distillation for heterogeneous federated learning[C]//International conference on machine learning. PMLR, 2021: 12878-12889.
>
> [5]Yoon T, Shin S, Hwang S J, et al. Fedmix: Approximation of mixup under mean augmented federated learning[J]. arXiv preprint arXiv:2107.00233, 2021.
>
> **Q3. In Section 4.3, each client uploads the distribution parameters to the server, but the paper states that ``the server does not have access to the sequence of the distribution parameters." This seems contradictory.**
>
> We apologize for the confusion. The server receives the distribution parameters uploaded by different clients. Then it aligns these distributions to identify the orthogonal ($\mathcal{I} _ o$) and parallel sets ($\mathcal{I} _ p$).
>
> **Q4. Errors and Typos**
>
> Thank you for your thorough review and for pointing out these grammar, formatting, and presentation issues. I have carefully addressed all the points in the revised manuscript.

---

### Official Review · Reviewer_mZWU · 2024-11-06

**Soundness:** 2
**Presentation:** 1
**Contribution:** 2
**Rating:** 3
**Confidence:** 3

**Summary:**

In this work, authors claim that a major challenge of federated learning is due to the entanglement of data distribution across clients. As a result, authors proposed a communication-efficient algorithm named FedDistr which decouples client data distributions into the different base distributions, and then the server merges the base distributions, finally a model trained over synthetic data generated following the base distributions. Authors perform numerical experiments to compare with several baseline federated learning algorithms and demonstrate FedDistr obtains better utility, communication, privacy trade-off.

**Strengths:**

1. Both communication cost and heterogeneity are important challenges in federated learning;
2. Authors verify the efficacy of the proposed algorithm through numerical experiments;
3. The proposed algorithm saves communication cost compared to baseline federated learning algorithms with little utility loss.

**Weaknesses:**

1. The paper is not well-motivated. In federated learning, heterogeneity is a more common term compared to the so-call disentanglement. Furthermore, in the first paragraph, authors claims 'There is a consensus that this inefficiency stems from the entanglement of data distribution across clients', can you provide references to this claim of 'consensus'. In fact, in the later part of the introduction, authors show that it is the disentanglement where classical FL algorithms like FedAvg does not perform well.
2. The proposed algorithm is not clearly introduced. Both Algorithm 1 and Section 4 only covers the parts until 'Activate Distribution Alignment', it seems there are steps after this in Figure 3 where clients generate data locally and train locally? Please explain more.
3. The claim of privacy preserving lacks rigorous discussion. How do you guarantee that the proposed algorithm does not leak user privacy. Roughly, the propose algorithm transfer cluster center of local datasets (in latent space) to the server, will this leak sensitive user information?

**Questions:**

Please see the weakness above and also the questions below:

1. How to do you plot figure 5? do you add noise somewhere to guarantee DP?
2. It seems the final training is performed over the so-called base distributions, in case of classification, is the learned model useful for the original client dataset?

---

> ### Author Response · Authors · 2024-12-04
>
> **Q1. In federated learning, heterogeneity is a more common term compared to the so-call disentanglement.**
>
> We use $\epsilon$-entangled to represent the extent of heterogeneity among different clients, as defined in Definition 2 of the main text. Specifically, when $\epsilon = 0$, it corresponds to the disentangled case, where the data distributions of different clients are entirely distinct. Furthermore, Theorem 2 establishes the relationship between utility loss and the entangled extent $\epsilon$.
>
> **Q2. Can you provide references to this claim of 'consensus'?**
>
> In distributed systems, tightly disentangled tasks exhibit greater efficiency[1], as complex tasks can be decomposed into simpler subtasks that can be assigned to multiple clients for parallel execution. For instance, in a medical imaging diagnosis task, if different clients (e.g., hospitals) naturally have a disentangled focus on specific sub-populations (e.g., one hospital deals more with elderly patients, while another focuses on pediatric cases), this inherent disentanglement enables each client to contribute highly specialized and complementary knowledge to the global model. Such complementary contributions reduce redundancy and conflict during aggregation, allowing the server to achieve efficient convergence with fewer communication rounds compared to scenarios with entangled and homogeneous data distributions.
>
> Existing algorithms such as FedAvg fail to actively leverage such disentanglement. Their reliance on simple aggregation mechanisms like weighted addition overlooks the underlying structure of the data distributions, leading to suboptimal performance, especially in heterogeneous settings. Theoretical results (Theorem 2) demonstrate that when the server actively leverages disentanglement, it is possible to achieve global model utility with only one round of communication while maintaining a tolerable utility loss. This highlights the potential for significant efficiency improvements under disentangled data distributions.
>
> [1]Coulouris G F, Dollimore J, Kindberg T. Distributed systems: concepts and design[M]. pearson education, 2005.
>
> **Q3. The proposed algorithm is not clearly introduced. It seems there are steps after this in Figure 3 where clients generate data locally and train locally.**
>
> We apologize for the confusion. Briefly, after the server actively aligns the uploaded base distributions, each client receives the distributed base distributions $v=[v_p,v_o]$ from server. For each base distribution embedding $v_o^i \in v_o$, clients use the latent diffusion model to generate data $\widehat{x} _ i= \lbrace \widehat{x} _ {i,j} | j\in [n_i] \rbrace$ following the base distribution.
> $$\widehat{x} _ {i,j} = D(f(z _ {T}, T, [p, v_o^i])), j \in [n_i] \text{, for } i \in [|v_0|],$$
> where decoder $D$ is for the inverse mapping of encoder $E$, $f$ is the denoising process, $T$ is the number of diffusion steps, and $[p, v_i]$ is the conditional prompt corresponding to base distribution $i$.
>
> Using the synthetic data, the clients locally train their downstream task model.
> $$\mathcal{F} _ {k}^{*} = \text{argmin} _ {\mathcal{F} _ {k}} \mathbf{E} _ {i\in [|v_0|]} \mathcal{L} _ {CE} (\mathcal{F} _ {k}(\widehat{x}_{i,j}), \widehat{y}_i) \text{, for } k \in [K],$$
> where $\mathcal{F} _ k$ is the local downstream model of client $k$, $\mathcal{L} _ {CE}$ is the cross entropy loss, $\{\widehat{x} _ {i,j} | j \in [n_i] \}$ is the generated data corresponding to base distribution $i$.
>
> **Q4. The claim of privacy preserving lacks rigorous discussion.**
>
> Please refer to Q1 of Reviewer n8vX for further details.
>
> **Q5. It seems the final training is performed over the so-called base distributions, in case of classification, is the learned model useful for the original client dataset?**
>
> The learned model is indeed useful for the original client dataset. This is because our work focuses on achieving global optimal [2,3], which ensures the model optimality across all clients collectively.
>
> [2]Li T, Sahu A K, Zaheer M, et al. Federated optimization in heterogeneous networks[J]. Proceedings of Machine learning and systems, 2020, 2: 429-450.
>
> [3]Li Q, He B, Song D. Model-contrastive federated learning[C]//Proceedings of the IEEE/CVF conference on computer vision and pattern recognition. 2021: 10713-10722.

---

### Note · Authors · 2024-12-24

I have read and agree with the venue's withdrawal policy on behalf of myself and my co-authors.